# Oncological Outcomes of Primary vs. Salvage OPHL Type II: A Systematic Review

**DOI:** 10.3390/ijerph19031837

**Published:** 2022-02-06

**Authors:** Carmelo Saraniti, Barbara Verro, Francesco Ciodaro, Francesco Galletti

**Affiliations:** 1ENT Clinic, Department of Biomedicine, Neuroscience and Advanced Diagnostic, University of Palermo, 90127 Palermo, Italy; carmelo.saraniti@unipa.it; 2Division for Otorhinolaryngology, Department of Adult and Development Age Human Pathology “Gaetano Barresi”, University of Messina, 98125 Messina, Italy; dottfciodaro@alice.it (F.C.); fgalletti@unime.it (F.G.)

**Keywords:** otolaryngology, head and neck, laryngectomy, surgical oncology, salvage therapy

## Abstract

*Background:* Open partial horizontal laryngectomy type II (OPHL type II) has two main aims: oncological radicality and laryngeal preservation. The aim of this review is to define and emphasize the oncological efficacy of OPHL type II, both as primary and salvage surgery, by analyzing the latest literature. *Methods:* The research was carried out on Pubmed, Scopus and Web of Science databases, by using strict keywords. Oncological outcomes were evaluated by the following parameters: overall survival, disease-specific survival, disease-free survival, local control, laryngeal preservation, local recurrence. *Results:* The review included 19 articles divided into three groups: (1) primary OPHL type II, (2) salvage OPHL type II, (3) adjuvant radiotherapy after primary OPHL type II. The articles showed excellent results as far as oncological radicality and organ preservation. *Conclusions:* This review demonstrated that OPHL type II is useful to obtain oncological radicality both as primary surgery and salvage surgery. Nevertheless, the only criterion that determined the positive outcome and efficacy of this technique is the strict selection of patient and tumor.

## 1. Introduction

Supracricoid laryngectomy, or open partial horizontal laryngectomy type II (OPHL type II), introduced in the 1950s [1] and modified by Piquet in 1974 [2], has two main aims: oncological radicality and organ preservation [3]. This surgery technique enabled researchers to limit total laryngectomy surgery and its consequences: permanent tracheostoma and loss of the natural voice [4]. In fact, in 2018, the American Society of Clinical Oncology (ASCO) recommended total laringectomy only for extensive T3 and T4a lesions to ensure better survival rate and recommended chemoradiotherapy or OPHL for locally advanced disease to ensure the greatest possible organ preservation and minimal functional impairment [5]. Particularly, T2 and selected T3 glottic and supraglottic cancers [6] are amenable with OPHL type II by removal of thyroid cartilage, true vocal folds and false vocal folds, Morgagni’s sinus, pre-epiglottic space and paraglottic space. Therefore, the only preserved structures are: cricoid cartilage, one or both arytenoid cartilages, hyoid bone, and, sometimes, suprahyoid epiglottis. According to the 2014 classification by the European Laryngological Society (ELS) [7], OPHL type II can be divided into two subtypes: type IIa if suprahyoid epiglottis is preserved, so crico-hyoido-epiglottopexy (CHEP) is performed; type IIb if suprahyoid epiglottis is removed and crico-hyoidopexy (CHP) [8] is performed. The crico-arytenoid unit (CAU), composed of the crico-arytenoid joint and the underlying hemicricoid plate, guarantees the satisfactory functional outcome of this surgery; for this reason, it is fundamental to preserve at least one mobile arytenoid cartilage. Furthermore, Calearo and Bignardi, with their histological studies on the larynx and laryngeal carcinoma, have demonstrated the importance of the arytenoid for oncological radicality: in fact, thanks to the fibrous ligament on its anterior aspect, it acts as a barrier hindering the extension of the tumor to the underlying laryngeal structures [9].

Contraindications to OPHL type II are: (1) extension of the tumor to both arytenoids, to the crico-arytenoid unit or to the posterior commissure, (2) invasion of the hyoid bone, (3) extension of the tumor to the cricoid cartilage, and (4) extralaryngeal spread.

Thus, OPHL type II also serves an important role in salvage surgery after the failure of radiotherapy or TLM (transoral laser microsurgery) [10,11], reducing the use of total laryngectomy. Therefore, this partial surgery enables researchers to obtain reliable, functional and oncological results both as primary surgery and salvage surgery [12,13]. 

However, selection of the patients eligible for OPHL type II [10] is determined by considering localization and extension of the tumor, as well as the health and psychosocial status of the patient.

The aim of this review is to define and emphasize the oncological efficacy of OPHL type II, both as primary surgery and salvage surgery, as far as overall survival (OS), disease-specific survival (DSS), disease-free survival (DFS), local control (LC), laryngeal preservation (LP) and local recurrence (LR) are concerned, by analyzing the latest literature on this topic. 

## 2. Materials and Methods

### 2.1. Search Methodology

The selection of bibliography was carried out on Pubmed, Scopus and Web of Science databases, by using the following keywords: *supracricoid laryngectomy* or *open partial laryngectomy type II* or *OPHL type II* and *oncologic outcome*. Furthermore, some articles were chosen from a bibliography of selected studies. Therefore, two independent authors (BV and CS) selected the articles: a first selection was made by reading titles and abstracts. Afterwards, the selected articles were read entirely in order to include only those that fulfilled the eligibility criteria in the study.

### 2.2. Eligibility Criteria

The eligibility criteria were: (1) studies that analyzed oncological outcomes exclusively of OPHL type II, (2) at least 30 patients included in the study, (3) studies that calculated at least two of the following parameters: OS, DFS, DSS, LC, LP, LR, (4) median or mean follow-up period of at least 36 months, (5) studies published from 2000 onwards. 

Exclusion criteria were: (1) reviews, editorials, opinions or case reports, (2) studies that did not distinguish OS, DFS, DSS, LC, LP, LR in primary and salvage OPHL type II, (3) articles that were not written in English or Italian, (4) studies in which patients had undergone adjuvant chemotherapy. 

### 2.3. Data Analysis

The following information was selected from chosen articles: authors, year of publication, number of included patients, follow-up duration (months), OS, DFS, DSS, LC, LP, LR. Data were collected on data spreadsheet using Microsoft Excel (version 16.47.1).OS is the time between surgery and death by any cause or last follow-up; DSS is the interval time between surgery and death from the disease; DFS is the time from surgery to tumor recurrence; LC is the length of time from surgery to relapse on the primary tumor site; LP is the interval time between OPHL to total laryngectomy; LR is the time between surgery and the first local recurrence [14,15].

## 3. Results

### 3.1. Selection and Classification of Studies

Figure 1 shows the process of selection of the studies [16]. Overall, 145 articles were selected from the systematic research. The first step included the elimination of doubles (n° 23) and, secondly, the exclusion of unrelated articles, taking into account title, abstract or language criteria (n° 90). Subsequently, 48 full studies were read independently by the two authors (BV and CS) and assessed, taking into account the eligibility criteria, including only 19 articles in the review.

Afterwards, the selected articles were analyzed and grouped into three different categories: (1) primary OPHL type II, (2) salvage OPHL type II, (3) adjuvant radiotherapy after primary OPHL type II.

### 3.2. Primary OPHL Type II

Six primary OPHL type II [3,4,6,11,17,18] articles were selected and assessed (Table 1). Nearly all the studies included 5-year OS, corresponding to about 80% [3,4,11,17,18], except for the study by Sánchez-Cuadrado et al. [6] with a 60% 5-year OS in a sample of 41 patients. Primary OPHL type II was also efficient and resolutive considering 5-year DSS (76.7% [4] and 82.4% [18]) and LC (95.6% [3], 80% [6] and 93.94% [17]). Organ preservation, which represents the main goal of this surgery, was guaranteed by more than 85% of patients [3,6,11]. 

### 3.3. Salvage OPHL Type II

This group included five articles [10,11,19,20,21] (Table 2). Three studies [10,20,21] demonstrated that salvage OPHL type II guaranteed over 80% of 5-year overall survival. However, Deganello et al. [16] reported a 5-year OS of 60%. The 5-year LP was guaranteed in more than 90% of patients [10,19]: the use of total laryngectomy was, therefore, avoided in a really high percentage of patients. Pellini et al. [20] also reported excellent results in 5-year DFS (95.5%) in a sample of 70 patients; Bertolin et al. [21] obtained 86% of 5-year DFS in a sample of 50 patients.

### 3.4. Adjuvant Radiotherapy after Primary OPHL Type II 

Eight articles assessed the oncological outcome of OPHL type II combined with adjuvant radiotherapy [8,13,22,23,24,25,26,27] (Table 3). Overall, adjuvant RT proved to be adequate for: positive resection margin, thyroid cartilage invasion (stage T4a), positive neck nodes with extracapsular invasion, multiple nodal metastases, following the National Comprehensive Cancer Network (NCCN) [28] guidelines. The majority of the studies reported 5-year OS and DSS higher than 80%. Rizzotto et al. [24] obtained 5-year OS and DFS in 95.6% and 90.9% of patients respectively. Basaran et al. [13] divided patients into two groups: (1) both arytenoids preserved (BASCL) and (2) one arytenoid preserved (OASCL). Overall, no statistically significant differences were detected in the two groups in terms of oncological outcome.

## 4. Discussion

Laryngeal carcinoma accounts for about 2% of all cancers in the world [29]. For early and locally advanced laryngeal cancers, several therapeutic strategies are available: TLM, radiotherapy, chemoradiotherapy and open laryngeal organ preservation surgery (OLOPS) [30,31,32,33].

In particular, supracricoid laryngectomy is an open partial laryngeal surgery that has two main goals: radical excision of laryngeal cancer and preservation of functions (swallowing, phonation and breathing). In fact, while removing portions of the larynx, this surgery, in its reconstructive phase, allows for restoring the physiological crossway between the digestive and respiratory tract.

OPHL type II is, therefore, recommended for selected supraglottic and glottic cancers [6], in which it manages to guarantee good oncological and functional outcomes, thus, limiting the use of both primary and salvage total laryngectomy. In particular, in 2018, the American Society of Clinical Oncology (ASCO) recommended: TLM or radiotherapy for T1 and T2 laryngeal cancers with the goal of preserving the larynx; OPHL or chemoradiotherapy for locally advanced disease (T3, T4 laryngeal cancers), in order to achieve the greatest possible organ preservation with minimal functional impairment; total laryngectomy for extensive T3 and T4a lesions for a better survival rate [4]. 

In regard to chemotherapy for locally advanced disease, in the same paper, ASCO stated that concurrent chemoradiotherapy (CRT) guarantees satisfactory results in terms of laryngeal preservation compared to RT alone, although with high in-field toxicity [4]. Furthermore, ASCO advised against induction chemotherapy before laryngeal preservation surgery, even though Luna-Ortiz et al. [34] proved that induction chemotherapy allowed one to perform OPHL type II even in the case of arytenoid fixation (which is a contraindication to this surgical technique) determining the recovery of motility, without any impairment of DFS and/or OS.

As for total laryngectomy, it represents a real amputation of an organ that strongly characterizes the individual and that is essential for breathing, swallowing and speaking. Furthermore, this surgery involves the creation of a permanent tracheostoma which creates an important and significant impact on the patient’s psychology and overall quality of life [35]. Weinsten et al. [36] compared the quality of life using the SF-36 general health status system and the V-RQOL (Voice-Related Quality Of Life) test [37,38], showing significantly better results in the OPHL type II group compared to the total laryngectomy group.

However, patient selection is mandatory to provide the best treatment in both oncological and functional terms [9]. This selection is based not only on the characteristics of the tumor, i.e., localization and local-regional extension, but also on the health and psychosocial state of the patient himself. Therefore, from the oncologic point of view, OPHL type II is indicated in the case of T2 and selected T3 glottic and supraglottic cancers. The most important factors based on the patient’s characteristics, on the other hand, are: age, intellectual abilities and pulmonary function [21]. In particular, the patient’s age parameter (cut-off 70 years) [31] has always been the subject of discussion and debate. According to some authors, in fact, advanced age does not represent a contraindication to the intervention of OPHL type II [6] due to the difference between biological and chronological age [31]. According to other authors, however, advanced age correlates with a worse functional and clinical outcome [39]. Furthermore, Lucioni et al. [40] indicated young age as a negative prognostic factor, probably due to the increased aggressiveness of the tumor. Crosetti et al. identified other exclusion criteria: severe metabolic diseases, neurological and/or pulmonary diseases that compromise the ability to swallow and expectorate, severe heart diseases [41].

Finally, the choice of the best therapeutic strategy should always be agreed with the patient and with the family members too, highlighting the importance of involving the family in the final decision, making them aware of the therapeutic options and the advantages and disadvantages of each [10].

### 4.1. Primary OPHL Type II

Primary OPHL type II is recommended in cases of T2 and selected T3 glottic and supraglottic cancers. In particular, the American Society of Clinical Oncology suggests OPHL type II as first choice for T2 tumors because it achieves better oncological outcomes than primary RT [42]. ASCO also clarified that laryngeal preservation surgery should always be preferred to RT for T1 and T2 laryngeal cancer but underlined that TLM is not always feasible depending on these key points: endoscopic tumor exposure, endoscopic technique safety and surgeon’s experience [4]. In fact, performing surgery with the awareness of not being radical (close or positive resection margins) and, therefore, of having to do post-operative RT, is not an acceptable therapeutic option in any way. For this reason, in these cases, OPHL type II represents a valid alternative to the TLM, before RT.

Moreover, surgery is preferred to RT because of toxicity, odynophagia, hoarseness and thick salivary secretions and, above all, it could relate to a high risk of dysphagia and aspiration. The selected articles showed the efficacy of primary OPHL type II as far as 5-year OS, DSS, DFS, LP, LC and LR are concerned. In their study, Page et al. [17] found some factors that relates to a statistically significant risk of local recurrence: age, lymph node positivity (N+), positive resection margin, other synchronous cancer. Furthermore, in their study, they underlined that the main risk factor for local recurrence (and, therefore, a negative prognostic factor) is the positive or close resection margins (healthy tissue-carcinoma distance <1 mm), especially the inferior margin [43]. For this reason, to guarantee a safe surgery, Page et al. stated that resection margins had to be superior to 1 mm and subglottic extension inferior to 10 mm. 

### 4.2. Salvage OPHL Type II

Taking into consideration the choice and the clinical conditions of the patient, as well as the care center, selected glottic and supraglottic carcinomas can be treated with primary radiotherapy [19]. In fact, RT guarantees the preservation of laryngeal anatomical structures and, therefore, a better functional result. Furthermore, in some cases, a tracheostomy is not necessary and, consequently, has a lower impact on the patient’s quality of life. However, this treatment has a recurrence rate between 5 and 30%, apart from the above-cited side effects. Atallah et al. found several factors that correlate to low local control rate in the case of RT for T2 glottic tumors: male sex, degree of tumor differentiation, administration modality of radiotherapy (total dose and fractionation of administration), the extension of the tumor to a subglottic level or the anterior commissure. In particular, in the latter case, the high risk of tumor recurrence or persistence is related to diagnostic and therapeutic limitations: (1) the difficulty of evaluating the possible involvement of the thyroid and cricoid cartilages and of the cricothyroid membrane and (2) the difficulty of irradiating this laryngeal region [26]. Thus, in these cases, OPHL type II is performed with two goals: safe tumor clearance and preservation of laryngeal functions [19]. However, it is important to underline that salvage surgery correlates to a high risk of complications (chondritis, salivary fistula, rupture of pexy) related to previous laryngeal irradiation. In fact, radiotherapy is responsible for a slow healing process with a delay in tracheal decannulation and recovery of swallowing function [44]. Nevertheless, the risk-benefit ratio in salvage OPHL type II is positive. Indeed, a valid disease local control emerged with OS, LP, and LC values being, on average, slightly lower than primary OPHL type II from the analysis of the selected articles. Furthermore, Pellini et al. [20] underlined that the recurrence after RT had a different diffusion pathway compared to virgin neoplasia, with tumor foci also being far from the initial site of the tumor. This concept still emphasizes the importance of a wide resection with safe surgical margins and frozen sections of the resection margins, although the evaluation is more problematic in tissues that have been previously radio-treated [11]. Nevertheless, in order to obtain satisfying oncological outcomes, the strict selection of the patients that can undergo partial laryngectomy is fundamental: therefore, it is not possible to consider OPHL type II as the standard salvage therapy [21]. 

### 4.3. Adjuvant Radiotherapy after Primary OPHL Type II 

In different studies, OPHL type II was associated with adjuvant RT. As a consequence, in order to avoid bias on the results, these articles were assessed separately from those that calculated the oncological outcomes after surgery (OPHL type II). All the studies included in the review showed satisfying results, with 5-year OS, DSS, LP and LC superior to 85%, on average. These results confirm what was stated by Atallah et al. [26]: combined therapy (surgery and adjuvant RT) gives a better local control than surgery or radiotherapy alone.

However, if post-operative RT guarantees improved oncological results, on the other hand, the risk of significant impairment of the functions of the residual larynx increases [19].

### 4.4. The Old Discussion: Laryngeal Preservation vs. Functional Preservation

The Achilles’ heel of OPHL, especially OPHL type II and type III, is the functional result in terms of voice and swallowing. In fact, these two functions are closely linked and interdependent: both depend on the ability of arytenoid(s) to perform the sphincter function. This depends more on the base of the tongue and the mucous thickness of neoglottis and arytenoids than on the motility and number of residual arytenoids or on the presence or absence of the epiglottis. Several studies have analyzed voice results in these patients and found a moderate to severe alteration in the quality of the voice [45]. However, as Schindler et al. stated, the voice, although qualitatively poor, is not perceived by the patient as a handicap characterizing the quality of life [46]. OPHL’s functional success is assessed on the basis of two main parameters: decannulation and nasogastric feeding-tube (NFT) removal. In this respect, the literature data were highly variable: Pinar et al. [8] reported a mean nasogastric tube removal of 11.43 days and a decannulation time of 16.79 days, On the contrary, Goncalves et al. [47] removed NFT and tracheal tube with a mean of 69 and 60 days, respectively. However, the decannulation rates were very high (over 85%), confirming good neoglottic patency [8,48]. In the case of impaired neoglottic patency, especially in patients with a poor cough reflex, tracheal aspiration (not only during feeding but constantly with saliva) can cause a serious complication, that is, aspiration pneumonia, with a lower than 20% incidence [47]. In the most serious cases, with frequent episodes of aspiration pneumonia, it is necessary to resort to total laryngectomy for a dysfunctional larynx with a very low rate (less than 2%) [49]. In fact, unfortunately, the preservation of the organ does not always correspond to function preservation, both in the case of surgery and RT. Therefore, in the choice of the best therapeutic approach for the patient with laryngeal carcinoma, it is mandatory to consider pros and cons of therapy and to determine, together with the patient, the best therapeutic strategy in terms of both oncology and function.

### 4.5. Study Limitation

First of all, despite the use of strict selection criteria, the included articles were heterogeneous in terms of number of patients and stage of larynx carcinoma. Indeed, it was impossible to analyze the oncological outcome considering the larynx carcinoma because the selected articles showed an overall result over the total number of treated patients. Furthermore, differentiating the oncological outcomes between OPHL type IIa and IIb was impossible in the review, because very few studies showed diversified results for these two categories.

## 5. Conclusions

The systematic analysis of the articles demonstrated that OPHL type II is useful to obtain oncological radicality. Indeed, this surgical technique proved to be efficient both as primary and salvage surgery. Furthermore, the study showed another important element: the strict selection of patients eligible for OPHL, based not only on the tumor’s characteristics but also on the health and psychosocial conditions of the patient himself.

## 6. Highlights


OPHL type II has two aims: oncological radicality and organ preservationOPHL type II ensures good oncological outcome both in primary and salvage surgeryOPHL type II performs better results in primary surgery than in salvage surgeryThe main criterion for positive outcomes is the strict selection of patient and tumor stage


## Figures and Tables

**Figure 1 ijerph-19-01837-f001:**
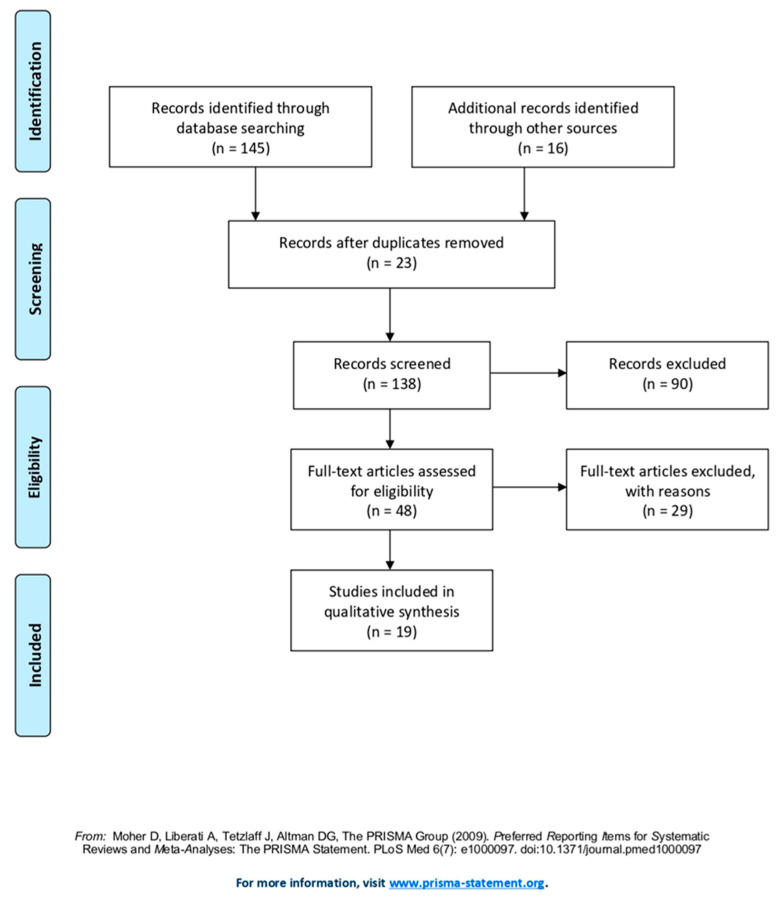
PRISMA 2009 Flow Diagram© of study selection process of literature (from [16]).

**Table 1 ijerph-19-01837-t001:** Primary OPHL type II: characteristics of included studies.

Authors (Year of Publication)	N° Patients	pT Treated	Follow-Up	OS	DSS	DFS	LP	LC
Karasalihoglu AR et al. (2004) [3]	68	T1-T4	62 months (median)	78.6% (5 years)	93.9% (5 years)	/	89.7% (5 years)	95.6% (5 years)
Sánchez-Cuadrado I et al. (2011) [6]	41	T1-T3	43 months (median)	69% (5 years)	81% (5 years)	/	85% (5 years)	80% (5 years)
Nakayama M et al. (2013) [11]	43	T1-T4	38 months (median)	81% (5 years) [salvage]—87% (5 years) [virgin]	/	/	94% (5 years) [salvage]—91% (5 years) [virgin]	/
Page C et al. (2013) [15]	291	T1-T3	56 months (mean)	80% (5 years)	/	/	/	93.94% (5 years)
Ozturk K et al. (2016) [5]	90	T1b—T2—selected T3	55 months (median)	80.4% (5 years)	/	76.7% (5 years)	/	/
Gong H et al. (2019) [16]	164	T1b—T2—selected T3	85 months (median)	86.9% (5 years)	87.6% (5 years)	82.4% (5 years)	/	/
Number of treated patients, stage of tumor (pT), period of follow up, overall survival (OS), disease-specific survival (DSS), disease-free survival (DFS), laryngeal preservation (LP), and local control (LC).								

**Table 2 ijerph-19-01837-t002:** Salvage OPHL type II: characteristics of included studies.

Authors (Year of Publication)	N° Patients	pT Treated	Follow-Up	OS	DSS	DFS	LP	LC
Deganello A et al. (2008) [19]	31	T1-T4	45 months (mean)	60% (5 years)	/	/	90% (5 years)	75% (5 years)
Pellini R et al. (2008) [20]	78	T1-T4	70 months (median)	81.8% (5 years)	/	95.5% (5 years)	/	/
Nakayama M et al. (2013) [11]	30	T1-T4	38 months (median)	81% (5 years) [salvage]—87% (5 years) [primary]	/	/	94% (5 years) [salvage]—91% (5 years) [primary]	/
Sperry SM et al. (2013) [10]	42	T1-T3	61 months (mean)	75% (5 years)	85% (5 years)	/	95% (5 years)	98% (5 years)
Bertolin A et al. (2020) [21]	50	T1-T4	50.1 months (mean)	82% (5 years)	88% (5 years)	86% (5 years)	/	/
Number of treated patients, stage of tumor (pT), period of follow up, overall survival (OS), disease-specific survival (DSS), disease-free survival (DFS), laryngeal preservation (LP), and local control (LC).								

**Table 3 ijerph-19-01837-t003:** Adjuvant radiotherapy after primary OPHL type II: characteristics of included studies.

Authors (Year of Publication)	N° Patients	pT Treated	Follow-Up	OS	DSS	DFS	LP	LC	LR	Criteria for Adjuvant RT
Gallo A et al. (2005) [20]	253	T1-T4	51.6 months (mean)	79.1% (5 years)	/	/	/	/	8.7% (5 years)	Positive resection margin, >N1, extracapsular spread
Pinar et al. (2012) [8]	56	T1-T4	58 months (median)	82.1% (5 years)	86.5% (5 years)	/	/	92.5% (5 years)	/	Positive resection margin, >N1, extracapsular spread, thyroid cartilage invasion
Topaloglu I et al. (2012) [23]	44	T2-T3	53.2 months (mean)	84.1% (5 years)	92.5% (5 years)	/	/	/	/	Positive resection margin, >N1, extracapsular spread, thyroid cartilage invasion
Rizzotto G et al. (2012) [24]	399	T1-T4	97 months (mean)	95.6% (5 years)	/	90.9% (5 years)	/	/	3.2% (5 years)	Positive resection margin, >N1, extracapsular spread
Mercante G et al. (2013) [25]	32	T3	47.3 months (median)	87.3% (5 years)	/	78.2% (5 years)	/	96.2% (5 years)	/	>N1, extracapsular spread, T4a
Basaran B et al. (2015) [13]	68	T2-T3	52.4 months (mean)	81.2% (5 years) [BASCL]—85% (5 years [OASCL] *p*-value 0.66	93% (5 years) [BASCL]—89.5% (5 years [OASCL] *p*-value 0.49	/	88.7% (5 years) [BASCL]—89.2% (5 years [OASCL] *p*-value 0.59	86.8% (5 years) [BASCL]—84.2% (5 years [OASCL] *p*-value 0.42	/	>N1, extracapsular spread
Atallah I et al. (2017) [26]	53	T1-T2	96 months (median)	93.7% (5 years)	95.6% (5 years)	87.7% (5 years)	/	/	11.3% (5 years)	Positive resection margin
Pescetto B et al. (2018) [27]	53	T1-T3	40.8 months (median)	86% (3 years)	95% (3 years)	80% (3 years)	/	/	/	/
Number of treated patients, stage of tumor (pT), period of follow up, overall survival (OS), disease-specific survival (DSS), disease-free survival (DFS), laryngeal preservation (LP), and local control (LC), local recurrence (LR), OASCL, both arytenoids preserved SCPL (BASCL), one arytenoid preserved SCPL (OASCL).										

## Data Availability

Publicly available datasets were analyzed in this study. This data can be found here: https://pubmed.ncbi.nlm.nih.gov/, accessed on 30 January 2021.

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
