# Peer review of "Oncological Outcomes of Primary vs. Salvage OPHL Type II: A Systematic Review"

_ijerph, 2022, doi:10.3390/ijerph19031837_

Round 1
Reviewer 1 Report
This manuscript is entitled: “Oncological outcomes of OPHL type II (primary vs salvage): a systematic review,” integrated that define and emphasize the oncological efficacy by Open Partial Horizontal Laryngectomy type II (OPHL type II). The authors search for both primary surgery and salvage surgery. Overall, the proposed research is of interest with good potential. The authors carried out detailed studies to prove the concept. There are some questions and suggestions that may need to solve as below:
- Authors should provide OPHL type II abbreviations in the abstract to make the entire paragraph easier to read.
- The data in the PRISMA 2009 Flow Diagram in Figure 1 seems to be taken from other sources, and please indicate the reference and citation.
- In the introduction section, the authors propose that the data were analyzed with various parameters such as overall survival (OS), disease-specific survival (DSS), disease-free survival (DFS), local control (LC), laryngeal preservation (LP), and local recurrence (LR), please give a detailed description of these parameters or cite relevant data and/or references to prove their importance.
- If the “/” in the table represents the lack of the data, it will be clearer to express it as n/a.
- Whether the data in the table are statistically relevant, such as p-value, standard deviation, etc., such data, if available, should be indicated on the table to provide more statistical confidence.
Author Response
COVER LETTER
Thanks for the suggestions and advice and comments that allowed us to make the article clearer and more complete.
- Authors should provide OPHL type II abbreviations in the abstract to make the entire paragraph easier to read.
As suggested by the reviewer, we have reported the abbreviation OPHL type II in the abstract.
- The data in the PRISMA 2009 Flow Diagram in Figure 1 seems to be taken from other sources, and please indicate the reference and citation.
With regard to the PRISMA 2009 Flow Diagram in Figure 1 we have indicated the corresponding reference, thus correcting the sequential numerical order of the references.
- In the introduction section, the authors propose that the data were analyzed with various parameters such as overall survival (OS), disease-specific survival (DSS), disease-free survival (DFS), local control (LC), laryngeal preservation (LP), and local recurrence (LR), please give a detailed description of these parameters or cite relevant data and/or references to prove their importance.
Thank you for the suggestion: in the section “materials and methods” we explained the meaning of each parameter examined with the appropriate references.
- If the “/” in the table represents the lack of the data, it will be clearer to express it as n/a.
In order to make the tables clearer and more comprehensible, we have replaced "/" with "n/a" as recommended by the reviewer.
- Whether the data in the table are statistically relevant, such as p-value, standard deviation, etc., such data, if available, should be indicated on the table to provide more statistical confidence.
We reviewed all the articles included in the study but the p-value could only be inserted in the article by Basaran et al. [13] where the results of each parameter were compared between the OPHL with both arytenoids preserved (BASCL) and those with one arytenoid preserved (OASCL).
- Lastly, the article was read and corrected again by a native English speaker.
We hope to have followed your directions correctly. We look forward to receiving your feedback.
Best regards
Barbara Verro
verrobarbara@gmail.com
ENT Clinic, Department of Biomedicine, Neurosciences and Advanced Diagnostic – University of Palermo, Palermo, Italy

Reviewer 2 Report
The authors present an interesting review study about the oncologic result using open partial larynx surgery. I would suggest to shorten the title by omitting the bracket. I am missing a few points in the discussion:
- laser surgery which is very suitable for T1,T2 and selected T3 glottic and supraglottic cancers of the larynx
- induction chemotherapy with the purpose of larynx preservation in cases T2 and T3 cases.
- The authors should look for functional analyses in the selected papers. However, this is not the main goal of this review but should be mentioned and discussed if there were any data reported, e.g. data on gastrostomy or tracheostomy rate/dependency etc. If there are any data reported then those should be discussed in relation to other treatment strategies. To my experience partial laryngectomy does often lead to severe functional impairment especially when it comes to silent aspiration. The old discussion about laryngeal preservation and functional preservation needs to be adressed more precisely.
Author Response
COVER LETTER
Thanks for the suggestions and comments that allowed us to make the article clearer and more complete.
- Laser surgery which is very suitable for T1, T2 and selected T3 glottic and supraglottic cancers of the larynx
Following the advice of the reviewer, in the discussions, we also investigated the role of transoral laser microsurgery by referring to the ASCO clinical practice guideline update (2018).
- Induction chemotherapy with the purpose of larynx preservation in cases T2 and T3 cases.
We have investigated the role of induction chemotherapy, with reference to a recent study (2022) which emphasizes its role before OPHL type II: in fact, it allows to perform this surgery even where it would have been contraindicated. On the contrary, as regards the role of induction chemotherapy, the ASCO is reluctant to use it.
- The authors should look for functional analyses in the selected papers. However, this is not the main goal of this review but should be mentioned and discussed if there were any data reported, e.g. data on gastrostomy or tracheostomy rate/dependency etc. If there are any data reported then those should be discussed in relation to other treatment strategies. To my experience partial laryngectomy does often lead to severe functional impairment especially when it comes to silent aspiration. The old discussion about laryngeal preservation and functional preservation needs to be addressed more precisely.
- As suggested by the reviewer, although it’s not the main goal of this review, for a more in-depth examination of OPHL type II, we have analyzed the functional results in terms of voice, swallowing and, therefore, decannulation rate and feeding nasogastric-tube removal rate.
- Lastly, the article was read and corrected again by a native English speaker.
We hope to have followed your directions correctly. We look forward to receiving your feedback.
Best regards
Barbara Verro
verrobarbara@gmail.com
ENT Clinic, Department of Biomedicine, Neurosciences and Advanced Diagnostic – University of Palermo, Palermo, Italy

Round 2
Reviewer 2 Report
Dear authors, thank you providing a revised and improved version of this manuscript.
I have to apologize but there are still some minor issues:
A) OPHL type II is therefore recommended for selected supraglottic and glottic cancers [6,32]
--> reference 32 is a case report and should be omitted here.
B) Laryngeal carcinoma accounts for about 2% of all cancers in the world [29]. For early and locally advanced laryngeal cancers, several therapeutic strategies are available: ELS, radiotherapy, chemoradiotherapy and Open Laryngeal Organ Preservation Surgery (OLOPS) [30-31].
--> References for Radiotherapy should be added
C) The authors should be consistent to TLM or ELS for laser surgery!
D) This concept still emphasizes the importance of a wide
resection with safe surgical margins and frozen sections of the resection margins, although the evaluation is more problematic in previously radiotreated tissues [11,44]. Nevertheless, in order to obtain satisfying oncological outcomes, the strict selection of the patients
that can undergo partial laryngectomy is fundamental: therefore, it is not possible to consider OPHL type II as the standard salvage therapy [21,45].
--> as this paragraph reports about salvage surgery references 44 and 45 are not proper selected and should be omitted, also to reduce self citations which is to be recommended in reviews in particularly.
Native english spell check is still required in particularly for the revised and newly integrated paragraphs.
